# Deep-Learning-Based Acoustic Metamaterial Design for Attenuating Structure-Borne Noise in Auditory Frequency Bands

**DOI:** 10.3390/ma16051879

**Published:** 2023-02-24

**Authors:** Ting-Wei Liu, Chun-Tat Chan, Rih-Teng Wu

**Affiliations:** 1Nikola Corporation, Phoenix, AZ 85040, USA; 2School of Mechanical Engineering, Purdue University, West Lafayette, IN 47907, USA; 3Department of Civil Engineering, National Taiwan University, Taipei 10617, Taiwan

**Keywords:** artificial intelligence, acoustic metamaterial, phononic crystal, noise control, deep learning, neural network, band gap

## Abstract

In engineering acoustics, the propagation of elastic flexural waves in plate and shell structures is a common transmission path of vibrations and structure-borne noises. Phononic metamaterials with a frequency band gap can effectively block elastic waves in certain frequency ranges, but often require a tedious trial-and-error design process. In recent years, deep neural networks (DNNs) have shown competence in solving various inverse problems. This study proposes a deep-learning-based workflow for phononic plate metamaterial design. The Mindlin plate formulation was used to expedite the forward calculations, and the neural network was trained for inverse design. We showed that, with only 360 sets of data for training and testing, the neural network attained a 2% error in achieving the target band gap, by optimizing five design parameters. The designed metamaterial plate showed a −1 dB/mm omnidirectional attenuation for flexural waves around 3 kHz.

## 1. Introduction

### 1.1. Motivation and Relevant Works

Noises exist everywhere in our daily lives. To name a few, they could be from nearby traffic, the cooling fans of electronic devices, other people’s conversations, or some operating machines. Depending on the sources, noises can have a wide variety of frequency spectra, and to human perception, not all noises in different frequency bands sound equally loud, even under the same pressure level measured by instruments. Figure 1a shows the equal loudness contours according to the ISO 226 standard.

Each contour indicates the same loudness in terms of “phons” with respect to a 1 kHz pure tone. For example, 60 phons means the same loudness of a 60 dB, 1 kHz tone, and 0 phons indicates the minimum threshold of hearing. In general, human ears are less sensitive to lower frequencies. Engineers usually adopt frequency filters to account for such sensitivity variation. For example, the A-weighting curve is the most-commonly used to assess the noise level [1]. The curve defined in the ANSI S1.42 standard is shown in Figure 1b, and the filtered level is denoted in dBA. It is noticed that dips occur in the contours around 3 kHz in Figure 1a (marked by the shaded interval), indicating the increase of the sensitivity of hearing, which is also reflected by the positive gain in the A-weighting curve (Figure 1b). Such a boost is ascribed to auditory canal resonance (inside our ears), and the frequency band is important for understanding spoken language (for us and for smart assistant devices). This is because the consonants are found in the frequency range of 2–4 kHz, and they play important roles in speech intelligibility.

Not only speech will be easily masked out in the presence of noises around 2–4 kHz, noise in that frequency band is also what we are most sensitive to. Therefore, addressing noise in this particular band is crucial. As such, in this study, we focused on the prevention of structure-borne noises in this particular frequency range, where they originate from either vibration sources attached to the structure or acoustic waves in the fluid domain coupled to the structure, and these noises can be mitigated by blocking the elastic wave propagation in the structure. Conventional noise control approaches may include large impedance mismatching, tuned mass dampers, or employing damping materials such as rubber [2,3] or foam [4,5,6]. The associated downsides include either the total weight of the structure would increase, not favorable for lightweight structures, or the damping materials could suffer from aging.

In the past couple of decades, periodic acoustic and elastic wave metamaterials, also known as phononic crystals, have emerged to be capable of manipulating elastic waves from frequency bands as low as 10 Hz for seismic waves [7,8,9,10], auditory bands [11,12], and ultrasonic bands [13,14,15,16,17,18,19], to GHz bands such as piezoelectric waves [20,21,22,23]. Not only the phononic crystals have shown competence in vibration and noise control, as well as signal filtering applications, they have also been reported to be able to arrest crack propagation [24,25], enhancing the fracture resistance and durability of structures. Soft dielectric elastomers, along with other tunable mechanisms that can reconfigure the phononic structure in real-time [15,26,27,28], have also been investigated. Particularly for structural plate wave control in auditory bands, a few studies based on local resonant structures have been proposed [29,30,31], to synthesize band gaps for flexural waves in thin plates. However, these existing studies aimed at frequencies below 500 Hz for vehicle noise and vibration control purposes, and also, the resonant components largely increase the weight of the structure. Alternatively, sheet metal with a periodic stamped pattern could be a practical approach to attenuate the acoustic noise in the frequency band of 2–4 kHz, and the design of such metamaterials was considered in this study.

Regarding the design methodology, the most-widely adopted approaches are still physics-driven combined with analytical and numerical software, usually through a trial-and-error process, to synthesize metamaterials achieving user-desirable dynamic properties such as phase and group velocities, wave polarization, and band gaps [23,32,33,34,35]. These methods highly depend on designers’ insights about the physics system to search for the solution to the design parameters, and they become intractable to conduct when a large number of geometric and material parameters are involved. To efficiently explore the design space, existing optimization-based approaches have been proposed for material and structural design problems. Notable examples include the level set method [36,37], the (bi-directional) evolutionary structural optimization (ESO) method [38,39], and the topology optimization with solid isotropic materials with penalization (SIMP) method [40,41]. However, these studies aimed to design materials satisfying the structural performance under external loads, and hence, the objective function was formulated with structural compliance in the physical space, not in a transformed space, such as the frequency–wavenumber spectrum, e.g., the dispersion curves. In terms of the design of acoustic metamaterials, common approaches such as the genetic algorithm (GA) [42,43,44,45] or topology optimization combined with conventional optimizers [46,47,48,49,50,51] have been used to determine the effective material properties, the refractive index, as well as the phononic frequency band gap. Although these GA or gradient-based optimizers work in a transformed space, i.e., the frequency domain, the GA-based approaches are quite computationally intensive due to their nature of population searching, while the gradient-based approaches suffer from the need for a good initial guess. Moreover, these methods require an iterative search each time when the target objective is changed. As a result, there is an urgent need to develop novel design methodologies for the acoustic metamaterial community.

In the past couple of years, deep learning started being used in phononic and photonic metamaterial designs [52,53,54,55,56,57,58]. By inputting frequency domain characteristics, the deep neural network (DNN) returns design parameters in the physical domain. Some of them used geometric dimensions for the design parameters [53], while some used direct images as the output [56]. For phononic band gap design, only bulk elastic waves in square lattices have been considered. Until now, these studies focused on the fundamental methodology, and there is still a gap between existing studies and the realistic engineering scenarios. Especially, given that plates and sheet metals are the most-commonly used structural components in various fields such as aerospace and the automotive industry, there are still yet studies investigating deep learning in the design of phononic band gap metamaterials for blocking plate waves. In this study, the Mindlin plate formulation [59] was employed to model the phononic elastic plate for AI-assisted metamaterial design. Compared with the Kirchhoff–Love plate theory, which is only valid for thin plates (with a plate thickness *t* much smaller that the wavelength Λ), the Mindlin model remains accurate for relatively thick plates as the shear deformation along the thickness is taken into account by analytical profiles. It has been shown to be able to accurately model relatively thick phononic plates with the plane wave expansion method [60]. For finite-element analysis, the Mindlin plate theory greatly reduces the computational cost compared with the full 3D elastodynamics formulation. This is particularly important for AI-based design, including deep learning, GA, and other algorithms, for which a large amount of forward calculations is required. The efficient formulation makes the AI-based design practical and competitive compared with conventional optimization methods.

### 1.2. Contribution and Scope

In this study, we present a deep-learning-based methodology to determine the optimal design of phononic metamaterial plates for attenuating structure-borne noise. By forming the band gap in the auditory frequencies, flexural waves in the frequency range generated by vibration sources are stopped from propagating in the metamaterial, further reducing the acoustic pressure disturbance caused by flexural waves and, ultimately, diminishing the environment noise level. We show that, with appropriate training, the DNN can return the optimal design parameters for the phononic metamaterial, based on the desired center frequency of the band gap, and bandwidth, as the input. The main contributions of this study are summarized as follows:To the best of the authors’ knowledge, this is the first study using deep learning in the design of phononic plates for synthesizing the flexural wave band gap.To the best understanding of the authors, this is the first study to employ the Mindlin plate formulation in modeling the phononic elastic plate for AI-assisted metamaterial design.Although this study aimed at a specific engineering scheme, the proposed design framework can be easily adapted to different circumstances. By demonstrating the explicit procedures from the initial performance requirements to the final results, this study fills the gap between existing studies and the application end, expediting the real applications in practice.

The remainder of the paper is organized as follows. Section 2 describes the inverse design problem discussed in this study, as well as the dataset generation details for deep learning. Section 3 presents the methodology and the DNN training details. Section 4 reports the training results and the related discussions. Section 5 evaluates the performance of the metamaterial design given by the DNN, via numerical experiments. The summary is given in Section 6.

## 2. Problem Statement

In this study, we propose a phononic plate wave metamaterial, as well as the design approach based on deep learning. The target was a frequency band gap centered at 3 kHz with a 60% relative bandwidth (normalized by the center frequency). The proposed metamaterial was made of sheet metal with a periodically machined slot pattern. To ensure the maximum isotropy of elastic waves propagating in the plate, we considered a pattern with a honeycomb lattice periodicity and a three-fold rotational (or cyclic, $C_{3}$) symmetric unit cell. Since the conservative acoustic system possesses time-reversal symmetry, the frequency (*f*)–wavevector (k) dispersion, namely f(k), is an even function in k-space, f(−k)=f(k). Together with the $C_{3}$ spatial symmetry, the *f*–k dispersion then acquires $C_{6}$ symmetry, i.e., identical wave properties repeat every $60^{\circ }$ in the azimuth angle of the wavevector k, which is the highest possible rotational symmetry in 2D periodic materials, ensuring equal attenuation in almost all directions. As a comparison, metamaterials with a square lattice pattern [29,30,31,56] can only have up to $C_{4}$ symmetry.

The candidate unit cell patterns were composed of a morphable geometry controlled by five design parameters, as shown in Figure 2a,b.

In Figure 2a, *a* indicates the lattice constant, i.e., the smallest pitch between adjacent repeated patterns, and $w_{1}$, $h_{1}$, $w_{2}$, and $h_{2}$ label the widths and heights of two distinct rectangular slots. The region to be removed was then obtained by the union of $C_{3}$ cyclic duplicates of the rectangular patterns. Figure 2b shows the machined pattern of the unit cell. The proposed rectangular slot design was for the ease of manufacturing, as it can be easily machined with, for example, water jet or laser cutting. Under different application scenarios, users can propose different shapes such as including fillets to avoid stress concentration, as long as the pattern possesses $C_{3}$ symmetry and has enough morphing latitude. With the candidate patterns decided, the remaining task was to determine the five design parameters for the desired band gap frequency range. A flowchart illustrating the procedures of this study is shown in Figure 3.

The band gap of the metamaterial with a given set of design parameters can be extracted from the *f*–k dispersion relationship, i.e., the phononic band structure, which can be calculated by using the finite-element (FE) methods. Such procedures to obtain the band gap from the design parameters are the *forward calculations*, as shown in the left of Figure 3. Next, a DNN was built to capture the relationships between the band gap and the design parameters. Batch forward calculations with combinations of design parameters and the resultant band gaps provided the dataset for the DNN training and testing. The DNN was trained to return the design parameters as the output, upon a given band gap frequency range as the input. The procedures are shown in the middle of Figure 3. The final stage was the *inverse design*, as illustrated in the right of Figure 3. The user enters the desired band gap, and the DNN suggests a set of design parameters. To confirm the validity of the parameters, we conducted the forward calculation with the returned parameters and examined if the band gap matched the input.

The remainder of this section focuses on the numerical model of the wave physics used in the forward calculations.

### 2.1. Theoretical Background

The dynamics of an isotropic homogeneous linearly elastic solid medium is described by the Navier equation:(1)$$\left(\lambda +2\mu \right)\nabla \nabla \cdot u-\mu \nabla^{2}u=\rho \frac{\partial^{2}u}{\partial t^{2}},$$
where λ and μ are the elastic Lamé constants, u the displacement vector, and ρ the mass density. The metamaterial we considered was made of steel, which had λ=115GPa, μ=76.9 GPa, and $\rho =7850\;\mathrm{kg}/m^{3}$. It has been shown that the dynamical behavior of an elastic wave propagating in phononic plates can be very well described by the Mindlin plate theory together with Bloch’s theorem [60]. As we only considered the flexural waves (i.e., the fundamental antisymmetric Lamb mode) with a wavelength greater that the plate thickness, employing Mindlin’s plate formulation (assuming a parabolic shear strain profile through the plate thickness) can greatly reduce the computational cost in FE analysis without a loss of accuracy (compared with the full 3D elastodynamics model). Therefore, during the network training and design phase, where a large amount of computations were required, we adopted the Mindlin formulation, and for the final evaluation stage (Section 5), the full 3D formulation was used to validate the design.

For waves in a periodic structure with lattice constant a, Bloch’s theorem [61] states that the wavefunction is an a-periodic function times the plan wave function $e^{ik\cdot x}$. For a 2D periodic structure, there are two linearly independent lattice vectors $a_{1}$ and $a_{2}$, as shown in Figure 4a.

For a wave with Bloch wavevector k propagating in the structure, one can focus on only the unit cell and apply the k-dependent periodic boundary conditions (also known as the Floquet periodic boundary conditions) on the three opposite pairs of boundaries:
(2a)$$u\left(x+a\right)=e^{ik\cdot a}u\left(x\right),$$
(2b)$$\sigma \left(x+a\right)=e^{ik\cdot a}\sigma \left(x\right),$$
where σ is the stress tensor.

Solving for the eigenfrequency with different ks gives the *f*–k dispersion relationship. For the periodic structure, f(k) is periodic in k-space, with the reciprocal lattice vectors $b_{1,2}$ satisfying $a_{i}\cdot b_{j}=2\pi \delta_{ij}$, where $\delta_{ij}$ is the Kronecker delta. Let $a_{1,2}$ and $b_{1,2}$ be the column vectors of matrices A and B, respectively, then from the above relation, we obtain $B=2\pi A^{-T}$ and $b_{1,2}$. The smallest period in k-space is known as the first Brillouin zone, as shown in Figure 4b. Considering the time-reversal and rotational symmetries, the triangular region bounded by the three high-symmetry points Γ, K, and M in k-space contains the complete information of f(k) and is known as the irreducible Brillouin zone. As the extreme values of the frequency of each band occur on the boundary of the irreducible Brillouin zone, it is sufficient to calculate f(k) along the path Γ-K-M-Γ, and the obtained spectrum is known as the band structure. In practice, this is obtained by solving the eigenfrequencies of the unit cell with Floquet periodic boundary conditions (Equation (2)) with all **k**s along the path in a batch parametric sweep. Note that, as can be seen from Figure 4b, the Γ-K segment happens to be the $k_{x}$-dispersion and Γ-M the $k_{y}$-dispersion (observing the $C_{6}$ symmetry of f(k)), while, along path K-M, both the azimuth angle and the length of the wavevector k simultaneously evolve.

The formation of a band gap in the phononic band structure is usually ascribed to two mechanisms, Bragg scattering [62] and local resonance [63,64,65]. The former occurs when integer multiples of the wavelength Λ=2π/k match twice the lattice spacing *d* between planes of scatters along the normal direction, nΛ=2dsinθ, which, in the ω-k dispersion curves, appears as the splitting of two counter-propagating waves on the Brillouin zone boundary. The band gap due to local resonance, on the other hand, is due to the coupling between the propagating wave and the locally resonant mode of the unit cell structure. In the band structure, it is shown as the repulsion between a flat band (the locally resonant mode, having nearly zero group velocity) and the propagating mode. For strongly modulated phononic structures, both mechanisms can take place, and the band gap is usually the product of the combined effect mixed with other factors. Due to the complexity, even though the band-gap-forming physics are well acknowledged, no analytical model can accurately predict the band gap frequency of an arbitrary phononic structure, but only numerical computations, let alone an inverse function for design purposes. That is when a systematic optimization method needs to be used.

### 2.2. Finite-Element Analysis and Data Generation

In general, for forward calculations generating the band gap information, all numerical methods capable of calculating the band structure from the design parameters can be used, including the plane wave expansion (PWE) method [20,66,67], the finite-difference time domain (FDTD) method [68,69,70,71], multiple scattering theory (MST) [72,73,74], transfer matrix [52,75], and the finite-element (FE) method [76,77,78]. The FE method can model a complicated geometry and strongly discontinuous material interfaces with a fast convergence speed and was adopted in this study. The previously described model calculating the phononic band structure was assembled and computed using the commercial FE package COMSOL Multiphysics. Quadratic mixed interpolation of tensorial components (MITC)-type Mindlin shell elements [79] with a maximum element size of a/16 were used to discretize and model the phononic plate. The plate thickness was fixed at t=0.4 mm in this study, which can be adapted or left as a design parameter for different application scenarios. The Floquet periodic boundary conditions were implemented using the built-in interface of the software. Alternatively, this can be realized by mapping variables from one boundary to the opposite one and constraining the displacement and reaction force following Equation (2) using the weak formulation [80]. The ARPACK eigensolver [81] was used to compute the eigenfrequencies *f*.

To prepare the dataset for DNN training, the range for each parameter needs to be defined. For the lattice constant *a*, there is no strict restriction aside from it has to be a positive length. To give a reasonable range, we refer to Bragg’s condition and took d=a. However, before calculating the *f*–k dispersion of the metamaterial, one would not have information about Λ for the target frequency. Nonetheless, since only an approximated range is required, a simple estimation based on the flexural wave dispersion in a uniform thin plate (i.e., without the phononic pattern) can be used [82]:(3)$$f\Lambda^{2}=\pi t\sqrt{\frac{E}{3\rho \left(1-\nu^{2}\right)}},$$
where E=200 GPa and ν=0.3 are the Young’s modulus and Poisson’s ratio of steel, which can be readily converted from the Lamé constants. Substituting in Λ=2a, t=0.4 mm, and the material constants yields a simple relationship, $fa^{2}\approx 1\;\mathrm{Hz}\cdot m^{2}$. For f=3 kHz, it gives a≈18 mm. Note that, in a strongly modulated phononic structure, the *f*–k dispersion could be vastly distinct from that in the uniform plate. Therefore, a wide range of *a* was considered for the data generation, 5mm<a<55mm. Furthermore, Equation (Equation 3) was derived from the Kirchhoff plate model for a quick estimation, which is valid for thin plates only. If the estimated Λ (or *a*) is comparable with the plate thickness *t*, one could use a more accurate model such as Mindlin’s [59] or the Rayleigh–Lamb frequency equation [82] for the estimation of *a*. For the rest the parameters, namely the dimensions of the rectangular slots, $w_{1},h_{1},w_{2},h_{2}$, the restrictions were simply to guarantee the pattern does not penetrate the hexagonal unit cell boundary, e.g., $h_{1,2}<a/\sqrt{3}$, $w_{1}+w_{2}<a/2$. Observing the parameter ranges and restraints, several values per parameter were adopted, resulting in a total number of 660 combinations of parameters. Among them, ill-defined shapes (such as rectangle corners penetrating the unit cell border or a rectangle intersecting with its $C_{3}$ duplicates, resulting in unconnected domains) were excluded from the dataset. In the end, 360 sets of parameters (i.e., distinct metamaterial patterns) were used to generate the band structures for DNN training and testing.

## 3. Methodology

In this section, the method of inverse design based on deep learning is discussed. A neural network was built and trained to predict the design parameters for the metamaterial to meet the desired band gap.

### 3.1. Data Preparation

As mentioned earlier, only the flexural mode in the plate has a significant contribution to the noise in the air domain. Thus, only the band gap for the flexural mode was considered. Due to large separation in the wave speeds of different modes, in general, extra effort is required to find a common band gap for all modes, which is unnecessary as the additional restraints would only make the metamaterial difficult to accommodated in various applications. In the following paragraph, a filtering strategy is presented to preserve only the flexural mode in the band structure for DNN training.

There are three fundamental (lowest order) modes of elastic waves in a homogeneous plate, namely the longitudinal mode (fundamental symmetric Lamb mode $S_{0}$), the flexural mode (fundamental antisymmetric Lamb mode $A_{0}$), and the fundamental shear horizontal mode ($SH_{0}$). For waves propagating in the *x*-direction (the thickness is along *z*), roughly speaking, these three modes have the particle displacement polarized in *x*, *z*, and *y*, respectively. For the considered phononic plate, due to the presence of the vertical cutting edges, the $S_{0}$ and $SH_{0}$ modes are mixed, while the $A_{0}$ mode is still decoupled from them (due to the mirror symmetry with respect to the mid-plane). For each eigenmode in the band structure, the *z*-polarization ratio (*z*-PR) can be evaluated to distinguish the flexural modes from the rest:(4)$$z-PR=\frac{1}{A_{\mathrm{cell}}}\int_{\mathrm{cell}}{\left(\frac{u\cdot \hat{z}}{∥u∥}\right)}^{2}d^{2}x,$$
which averages the *z*-polarization of a mode shape throughout the unit cell, where $A_{\mathrm{cell}}$ is the area of the unit cell (as the 2D shell formulation was used) and $\hat{z}$ is the unit vector along the *z*-direction. The integrand is the square of the directional cosine between u and the *z*-axis; therefore, it is always true that 0≤z-PR≤1. The $S_{0}$/$SH_{0}$ mixed modes ($A_{0}$ mode) have small (large) *z*-PR values. A threshold of 0.6 was used to filter out the $S_{0}$/$SH_{0}$ modes. Figure 5a,b plot a typical band structure before and after filtering, respectively. It was calculated using one of the parameter combinations from the dataset, with a=15 mm, $w_{1}=6$ mm, $h_{1}=0.75$ mm, $w_{2}=1.05$ mm, and $h_{2}=6$ mm.

The two branches with large slopes (implying fast wave speeds) in Figure 5a indicate the $S_{0}$ and $SH_{0}$ modes and were removed from the band structure in Figure 5b.

In the filtered band structure for flexural modes, it is seen that the band gap is located between the third and forth eigenfrequencies (counting from the lowest, for any k). This is generally true for the considered type of phononic plates. The band gap is then bounded by the minimum and the maximum of the third and forth eigenfrequencies, respectively. The upper and lower bounds of the band gap, $f_{+}$ and $f_{-}$, are indicated as the red and green bars in Figure 5b, respectively.

The band gap can also be expressed in terms of the center frequency $f_{c}$ and the normalized half-bandwidth δ, such that $f_{\pm }=\left(1\pm \delta \right)f_{c}$, or inversely from the $f_{\pm }$ obtained from the calculated band structure; they are written as
(5a)$$f_{c}=\frac{f_{+}+f_{-}}{2},$$
(5b)$$\delta =\frac{f_{+}-f_{-}}{2f_{c}}.$$

$\left(f_{c},\delta \right)$ were adopted as the input to the DNN for band gap information (instead of $f_{\pm }$), as they are physically more independent quantities. Figure 6 plots the distribution of $f_{c}$ and δ for the 360 metamaterial configurations from the generated dataset.

The distribution of the center frequencies of the band gap $f_{c}$ spans from below 1 to 14 kHz, and the half-bandwidth δ ranges from 12.5–30%.

### 3.2. Network Architecture

As shown in Figure 7, we implemented a fully connected neural network that consisted of one input layer, one output layer, and five hidden layers. The input layer contains the information of the user-desired band gap, which is represented by the center frequency $f_{c}$ and the normalized half-bandwidth δ. The hidden layers were composed of five layers of hidden neurons with the rectified linear unit (ReLU) activation function. Batch normalization (BatchNorm1d) layers were included in the first three hidden layers to accelerate the network training by reducing internal covariate shift. The number of neurons in the five hidden layers was 8, 32, 64, 32, and 10, respectively. Note that, in order to ensure a proper convergence, typically, the training of a DNN requires an observation in the early stage of the loss curves. Therefore, a preliminary work was conducted to determine the aforementioned activation functions and the loss functions by investigating how the loss decreases. Furthermore, the implemented network architecture was chosen according to the findings in [52,83]. The output layer then predicts the geometric dimension of the desired metamaterial unit cell, i.e., *a*, $w_{1}$, $h_{1}$, $w_{2}$, and $h_{2}$.

### 3.3. Network Training

The building and training of the DNN were performed using the open-source machine learning package PyTorch on Google Colaboratory clusters. There were 70% of the dataset (252 out of 360 sets) designated as the training group, whereas 21% (75/360) were assigned to the validation group. The remaining 9% (33/360) were used for testing. During the training process, 12 batches were chosen for each epoch. A total of 400 epochs were planned, with the learning rate dropping starting at 0.01. In terms of optimizer selection, Rectified Adam (RAdam) was adopted. It benefits from Adam’s quick convergence advantage, while also having good convergence results. The mean absolute error (MAE) of the predicted design parameters was used as the loss function for training, explicitly
(6)$$\mathrm{MAE}=\frac{1}{n}\sum_{j=1}^{n}\left|{\hat{y}}_{j}-y_{j}\right|,$$
where ${\hat{y}}_{j}$ and $y_{j}$ indicate the predicted and the ground-truth values of the five design parameters, respectively.

## 4. Neural Network Testing Results and Discussion

After being trained with 252 sets of data, the DNN was put under test with the 33 sets of data from the testing group. This section presents the testing results and some discussions.

### 4.1. Error Metrics for Parameters

While the MAE of the design parameters was used as the loss function during DNN training, it does not serve as a good indicator in interpreting the testing results since (1) the parameter MAE is in SI units, which is numerically small for the considered phononic plate, and does not provide physical significance without proper normalization and (2) it does not give errors for individual parameters, which may contain useful information and provide physical insights.

Three error metrics for the different design parameters are proposed for the testing, which are
(7)$$\mathrm{Error}=\left\{\begin{array}{cc} \frac{\hat{y}-y}{y},\; & \mathrm{for}\;a\;\mathrm{and}\;w_{1}\\ \frac{\hat{y}-y}{a\;/\;\sqrt{3}},\; & \mathrm{for}\;h_{1}\;\mathrm{and}\;h_{2}\\ \frac{\hat{y}-y}{a/3},\; & \mathrm{for}\;w_{2} \end{array}\right.\;.$$

Since the errors for individual parameters were not summed/averaged, not taking absolute values would allow us to better characterize the trend of the predicted parameters (if they are overall under- or over-estimated compared with the ground-truth). The three error metrics were normalized by different factors. For parameters *a* and $w_{1}$, which are the two largest geometric dimensions in the unit cell, the error was simply normalized by the ground-truth value.

For $h_{1}$, $h_{2}$, and $w_{2}$, the errors were normalized by their *maximum feasible values* instead of the ground-truth values. This was to avoid a meaningless large error indicator, which would fail to reflect the true level of inaccuracy in the presence of the small ground-truth values of these parameters (compared with *a*). For example, given a ground-truth value of $h_{1}=a/200$, a prediction ${\hat{h}}_{1}=a/100$ would yield a 100% error. However, practically, both parameters result in a narrow slit, or a thin crack, for which their difference is unobtrusive from the unit-cell-level point of view, and they have a negligible difference in terms of the band gap frequency. Accordingly, the errors in $h_{1,2}$ and $w_{2}$ were normalized by a/3 and $a/\sqrt{3}$, respectively.

In addition to the errors in design parameters, we also re-input the predicted parameters into the forward calculations (FE analysis) to retrieve the band gap for the metamaterial design given by the DNN ($\hat{f_{c}}$ and $\hat{\delta }$) and compared them with the input values ($f_{c}$ and δ). The errors were evaluated with the following metric: $\mathrm{Error}=|\hat{y}-y|/y$.

### 4.2. Network Prediction Results

Figure 8 shows the error distribution for each design parameter, to which the error metrics in Equation (Equation 7) were applied.

For parameters *a* and $w_{1}$, all predicted values lied within a 10% error, with a mean absolute value of 1.30%. The maximum error for parameter $h_{1}$ was 12% with a mean absolute value of 3.70%. For parameters $w_{2}$ and $h_{2}$, large errors were observed with a maximum absolute value of 20% and a mean absolute value of 7.17%. The overall mean absolute value for all errors in the five parameters (equally weighted) was 4.13%. It turned out that the DNN could accurately return predicted *a*, $w_{1}$, and $h_{1}$ that were close to the ground-truth values, but not for $w_{2}$ and $h_{2}$.

On the other hand, for the resultant band gap center frequency $f_{c}$ and half-bandwidth δ, the mean absolute errors were 2.26% and 1.75%, respectively, which are appreciably better than the errors in the design parameters and are satisfactory for practical applications.

To interpret such seemingly contradictory results, three case studies selected from the testing dataset (with the unit cells shown in Figure 9a–c), for which the DNN predictions (Figure 9d–f) contain representative results that help clarify the mentioned anomaly, are presented and discussed in detail in the following paragraphs. In short, this can be ascribed to two causes: (1) in some parameter ranges, the unit cell pattern is insensitive to some of the parameters, and (2) multiple designs lead to the same band gap frequency, while the DNN returns a parameter set distinct from the prepared dataset. In both circumstances, the DNN-predicted results showed large errors in the design parameters, while the resulting band gap still met the user input with a small error. The normalized mean absolute error, evaluated using Equation (Equation 7), of the five parameters and the normalized absolute errors in band gap for the three cases are listed in Table 1, and the shapes of the ground-truth unit cells from the dataset and the predicted ones are shown in Figure 9.

In Case 1, the parameter error (1.25%) was mainly contributed by $w_{2}$ and $h_{2}$ (the dimensions of the second rectangle slot), for which, as shown in Figure 9a,d, the two shapes are barely distinguishable, as the second rectangle is small compared with the unit cell. Furthermore, the predicted $w_{2}$ and $h_{2}$ were under- and over-estimated, respectively, resulting in almost identical areas of the slots. The calculated band structures were similar, and thus, the error in band gap frequencies were negligible (0.13% for $f_{c}$ and 0.98% for δ).

Case 2 tells a different story. Not only a large mean error in the parameters was observed (7.44%), but the predicted unit cell (Figure 9e) also visibly differed from the ground-truth unit cell (Figure 9b). Nonetheless, the resultant band gaps were almost identical (0.18% error in $f_{c}$ and 0.69% in δ). By checking the error of each parameter, it was noticed that the predicted *a* was over-estimated by 5.21%. The parameter *a* is considered as the most-important parameter controlling the band gap since it is proportional to the Bragg wavelength, for which a +5.21% error should result in a lower band gap frequency (larger Bragg wavelength). However, the parameters $h_{1}$ and $h_{2}$ were under-estimated by errors of −9.76% and −15.71%, respectively. This resulted in a faster speed of sound ($\mathrm{flexural}\;\mathrm{wave}\;\mathrm{speed}\propto {\left(\mathrm{bending}\;\mathrm{rigidity}/\mathrm{density}\right)}^{1\;/\;2}$), which compensated the lowered frequency, resulting in almost identical band gaps.

In general, a parameter set (five degrees of freedom (5 DOFs)) yielded a unique unit cell shape, and the band structure (f(k), infinitely many DOFs) was also unique (one-to-one, injective mapping). However, as we only extracted the band gap information (2 DOFs) from the band structure, such continuous mapping from 5-DOF to 2-DOF parametric spaces cannot be injective and is generally infinitely-many-to-one. In other words, there are infinitely many parameter combinations that could yield the same user input band gap. The fact that the DNN returned a parameter set different from the prepared ground-truth values, but matching the band gap, exactly showcases that the DNN captured the physics that link the band gap to the parameters and can give designs never seen in the dataset.

Figure 10a,b plot the band structures of the prepared ground-truth unit cell and the predicted one, respectively, for the discussed Case 2. It is seen that they have distinct (unique) f(k) dispersion curves (see, e.g., their third and fourth bands), but share identical band gap frequencies (labeled by the red and green bars).

Case 3 is among the worst cases found in the test results. Not only did it show a 6.47% error in the parameters, unlike in Cases 1 and 2, the error in $f_{c}$ in Case 3 reads as a 4.71% and 0.37% error in δ. The band structures of the prepared ground-truth unit cell and the predicted one are plotted in Figure 10c,d, respectively. The reason the DNN did not perform as good as in other cases could be due to the lack of training samples in the higher frequency range. As shown in Figure 6a, although $f_{c}$ spans up to 14 kHz in the dataset, the majority of the data lie below 6 kHz. In Case 3, the target $f_{c}$ is around 8 kHz, for which there are only a handful sets of training data near 8 kHz., thus the relatively poor performance. However, even for the worst case, the error in the band gap frequency should be acceptable in practical applications, considering 4.71% is even smaller than a semitone interval (i.e., between two neighboring keys on the piano, $2^{1\;/\;12}-1=5.95\%$).

To briefly summarize the test results, it is shown that, to evaluate the performance of the DNN, one should inspect the errors in the band gap ($f_{c}$ and δ), instead of comparing the predicted parameters with the prepared “ground-truth”, as there are infinitely many combinations of parameters that can yield the same user-desired band gap. For our trained DNN, the mean absolute errors of $f_{c}$ and δ were 2.26% and 1.75%, respectively, which are good for most applications, even though the DNN was trained with a limited amount of samples.

### 4.3. Baseline Reference

Four alternative machine learning algorithms, i.e., support vector regression (SVR), random forest regression (RFR), extreme gradient boosting (XGB), and K-nearest neighbors (KNN), were used as the baseline references to evaluate the performance of the proposed DNN employed in this study. The same training and testing datasets mentioned in Section 3.3 were adopted. In the baseline tests, the inputs were the desired band gap frequency and bandwidth, $\left(f_{c},\delta \right)$, and the outputs were the five design parameters $\left(a,w_{1},h_{1},w_{2},h_{2}\right)$. The five obtained design parameters returned were then sent back to the forward calculation, to calculate the band gap $\left({\hat{f}}_{c},\hat{\delta }\right)$ of the metamaterial design suggested by each baseline method. The performance of each algorithm was then evaluated by the error in $f_{c}$, i.e., $\left({\hat{f}}_{c}-f_{c}\right)/f_{c}$, and the error in δ, i.e., $(\hat{\delta }-\delta )/\delta$, as listed in Table 2. All four baseline methods underwent the fitting procedure using grid search to ensure a proper hyperparameter tuning. Furthermore, in order to predict the five parameters, a multi-output regressor was utilized if needed. The scoring for the grid search was the MAE, which was the same criterion used for DNN training. In SVR, the kernel in use was the radial basis function (RBF). Through several attempts in the grid search, the best parameter set was obtained for each machine learning method. The detailed hyperparameters of each model are reported in Table 3.

According to Table 2, SVR performed the worst among the five algorithms on both metrics. The DNN had the best accuracy in $f_{c}$ and a decent error in δ, which was slightly higher than that for RFR. However, the proposed DNN outperformed RFR by 1.78% in predicting $f_{c}$. In terms of the overall error rate, KNN had the second-best performance. Compared to the DNN, neither $f_{c}$ nor δ differed by more than 1%. XGB was marginally inferior to RFR in terms of both error metrics. This shows that, overall, the proposed DNN model achieved the best performance for band gap prediction.

Table 4 reports the training time, memory demand, prediction time, and number of parameters of the proposed approach, compared against the baseline methods. The values of time and memory were obtained by computing the mean out of five repeated trials, for each algorithm. Although the proposed DNN achieved the best performance when designing the unit cell, the DNN required the longest training time. Once the training finished, the proposed DNN achieved a prediction time of 0.0019(s) to complete one design query, which was superior to SVR, RFR, XGB, and KNN. Since the network training is usually conducted offline, the efficiency during the inference stage possesses greater interest in practice. With regard to the memory demand of the proposed approach, the instantaneous peak amount of RAM utilized during prediction was 0.0130(MB), which was lower than for SVR, RFR, XGB, and KNN as well. Notice that, due to the small size of our dataset, KNN had the first runner-up performance in terms of the prediction time and peak memory during inference when compared to the proposed DNN. However, it should be noted that the computation of KNN will scale up rapidly with the increasing number of training samples, since KNN needs to compute the distances between the design query and each training sample. This demonstrates the advantage of the proposed DNN approach over KNN. To ensure fairness, all the values in Table 4 are reported using the same CPU, which was the Intel Xeon Processor in 2.20 GHz with a single core, two threads, 2 MB L2 Cache and 12.7 GB RAM. The memory size of the trained DNN was 31(KB) only.

## 5. Validation of the Metamaterial Performance

So far, we have demonstrated that the trained DNN is capable of the inverse design of the phononic metamaterial that can achieve the desired band gap, In this section, we validate the performance of the obtained metamaterial design in terms of attenuating flexural waves via numerical transmission simulations.

### 5.1. Model Description

As mentioned in Section 1.1, noise in 2–4 kHz is most perceived by human ears. Therefore, the target band gap of the metamaterial was determined to be around 3 kHz with a 60% bandwidth (2.1–3.9 kHz). That is, $f_{c}=3$ kHz and δ=0.3 for the DNN input. The design parameters returned from the DNN were a=21.2 mm, $w_{1}=8.48$ mm, $h_{1}=1.58$ mm, $w_{2}=2.16$ mm, and $h_{2}=11.2$ mm. Two supercell (stacking of a finite number of unit cells) transmission FE models were built, one along Γ-K (with supercell length 8a=170 mm along the *x*-direction, equivalent to azimuth angles θ=nπ/6,n∈Z; see Figure 11a) and the other along Γ-M (with supercell length $4.5\sqrt{3}a=156$ mm in the *y*-direction, equivalent to azimuth angles $\theta =(n+\frac{1}{2})\pi /6,n\in Z$; see Figure 11b), respectively. A harmonic line load of unit strength along the *z*-direction $F_{z}=e^{i2\pi f\tau }$ (where τ indicates time) was applied on one side of the supercell. Perfectly matched layers (PMLs) with proper absorption wavelength were placed at the two terminals to eliminate unwanted reflections on the model boundaries. Periodic boundary conditions were set on the lateral boundary pairs, as shown in Figure 11.

Mixed hexahedral and pentahedral prism solid elements, using quadratic serendipity shape functions, with a maximum element size of a/16 (t/2) in the xy-plane (*z*-direction), were used for discretization and modeling. The 3D elastodynamic formulation was considered. Frequency responses from 50 Hz to 5 kHz with a step size of 50 Hz were computed.

### 5.2. Simulation Results

The transmission loss (TL) is defined by the ratio between the power (*P*) of the incident and the transmitted (out-going) waves, in decibels, as
(8)$$\mathrm{TL}=10\log_{10}\frac{P_{i}}{P_{o}}$$

In the FE models described previously, the out-going power can be easily obtained by integrating the mechanical energy flux I·n on the exit boundary (adjacent to the PML; see, e.g., the right side in Figure 11a), where n is the unit normal vector of the boundary, and in the mechanical energy flux, I=12Re(σ·v), where σ and v are the stress tensor and the particle velocity, respectively, where the cycle average was taken over a harmonic period.

However, the incident power cannot be obtained in a similar way on the opposite boundary since it would include the power of the waves reflected by the phononic structure. Instead, we created a reference model that had the same dimensions as those shown in Figure 11, but without any phononic pattern (a homogeneous plate), and the reference out-going power was used as the incident power in Equation (Equation 8). The transmission loss plots in the two directions (Γ-K and Γ-M) are shown in Figure 12a,b, respectively.

Both models showed significant attenuation from 2.1 to 4 kHz, and 180 dB and 150 dB transmission losses were found near the bottom of the curves for the Γ-K and Γ-M models, respectively. Note that TL=180 dB indicated that only $10^{-18}$ of the incident power was transmitted to the other side of the phononic structure, equivalent to a transmission coefficient of $10^{-9}$ in displacement, velocity, or acceleration. Recall that the two models have distinct lengths of the phononic barriers. If normalizing the transmission loss by the length of the phononic barrier, roughly 1 dB/mm of transmission loss was found in both directions. Given these two directions are only $30^{\circ }$ apart in azimuth angle and the transmission loss has the $C_{6}$ symmetric pattern, an omnidirectional 1 dB/mm peak transmission loss in the xy-plane was expected for the considered phononic metamaterial.

Figure 12c,d visualize the flexural wave attenuation by plotting the *z*-displacement amplitude $\left|u_{z}\right|$ in selected frequencies for both models. The color scheme utilizes dB scaling with the displacement amplitude of the reference model (homogeneous plate) as the 0 dB reference:(9)$$u_{z}-level=20\log_{10}\frac{\left|u_{z}\right|}{\left|u_{z,ref}\right|},$$
where the sampling locations for $u_{z,ref}$ are labeled in Figure 12c,d. The coefficient 20 (instead of 10) in Equation (Equation 9) was due to that energy being proportional to the square of the displacement amplitude. The exponentially decay in $u_{z}$ of approximately a −1 dB/mm attenuation in the band gap appeared in both cases, while outside the band gap, only minor attenuation (due to scattering and impedance mismatching between homogeneous and phononic plates) was found. These FE numerical experiments confirmed that the phononic metamaterial designed using the proposed deep learning approach can inhibit flexural wave propagation in the desired band gap with a peak attenuation of −1 dB/mm omnidirectionally.

## 6. Conclusions

In order to ease the process of phononic metamaterial design and to expedite the development cycle, we proposed an inverse design workflow incorporating deep learning. Compared with the conventional design approaches, such as trial-and-error, the gradient-based optimizer, or the genetic algorithm, a huge advantage of the proposed approach is that, once the deep neural network (DNN) is trained, it can be used multiple times (e.g., designing new metamaterials for different frequency ranges) without requesting any forward calculation (which usually involves costly computations, e.g., FE analysis). The considered phononic metamaterial aimed at blocking flexural wave propagation in the auditory bands by devising the band gap in the frequency–wavevector dispersion relationship. To generate the training dataset, the Mindlin plate formulation was employed in the FE forward calculations, which greatly reduced the computational cost. After the DNN training is finished, the user simply needs to input the center frequency and the bandwidth of the desired band gap, and the network will return a set of design parameters for the unit cell so that the metamaterial meets the desired band gap. For the metamaterial type considered in this study, we reported the average errors in the center frequency and the bandwidth to be 2.26% and 1.75%, respectively, and the baseline comparison indicated that the proposed DNN outperformed the other machine learning algorithms in achieving the target band gap. The results demonstrated that, for the human-sensitive auditory band (i.e., 2–4 kHz), the proposed metamaterial design approach achieved a −1 dB/mm omnidirectional attenuation for flexural waves around 3 kHz. It is also worth mentioning that the DNN returned decent predictions even with a limited amount of training samples.

With the promising results presented in this study, we expect that the proposed deep learning model can be adapted for metamaterial design with more parameters, including not only geometric dimensions, but other properties such as material selections. Furthermore, as the same band gap can be realized by more than one set of design parameters, multiple requirements in addition to the desired band gap can be considered at the same time, such as the minimum mass for lightweight structures or structural integrity for load-bearing structures, by formulating composite/joint objective functions. Future endeavors will be dedicated to the aforementioned aspects, as well as experimental validations of the proposed metamaterials.

## Figures and Tables

**Figure 1 materials-16-01879-f001:**
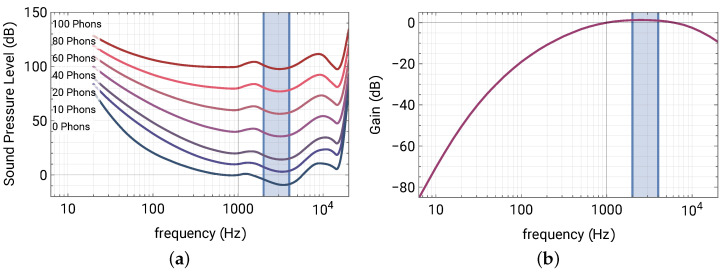
(**a**) Equal loudness contours (ISO 226). Dips around 2–4 kHz indicate the increase of hearing sensitivity to sound pressure. (**b**) The dBA weighting curve (ANSI S1.42) accounting for equal loudness.

**Figure 2 materials-16-01879-f002:**
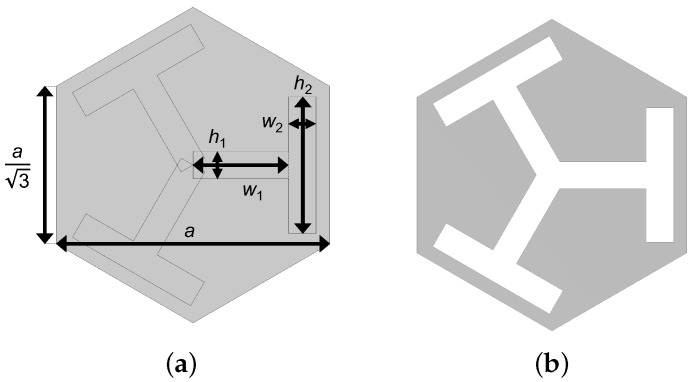
(**a**) Design parameters of the metamaterial unit cell. *a* indicates the lattice constant, i.e., the pitch between adjacent identical patterns, and $w_{1}$, $h_{1}$, $w_{2}$, and $h_{2}$ label the widths and heights of two distinct rectangular slots. The cut area was obtained by $C_{3}$ cyclic duplication of the rectangular slots. (**b**) A sketch showing the machined pattern of the unit cell.

**Figure 3 materials-16-01879-f003:**
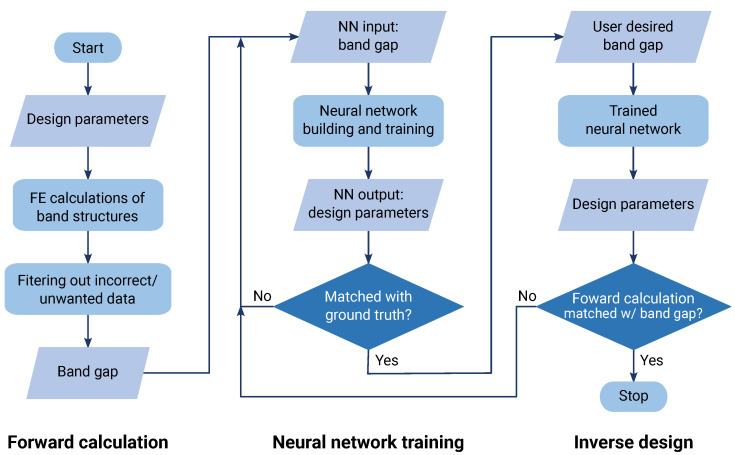
The proposed framework for the metamaterial design in this study.

**Figure 4 materials-16-01879-f004:**
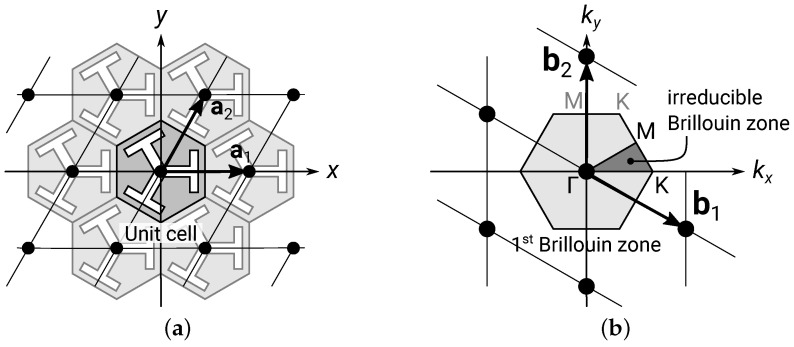
(**a**) An illustration of the real space pattern of the phononic lattice, where $a_{1,2}$ are the lattice vectors with $∥a_{1,2}∥=a$ the lattice constant. (**b**) The reciprocal space (k-space) periodicity. $b_{1,2}$ are the reciprocal lattice vectors, and the primitive unit cell in k-space is the first Brillouin zone. The area bounded by Γ-K-M-Γ is the irreducible zone observing all symmetries of the lattice.

**Figure 5 materials-16-01879-f005:**
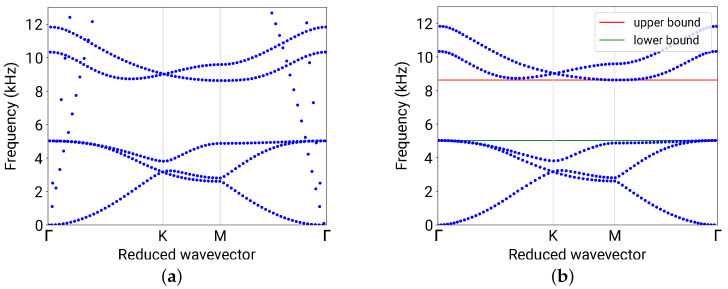
A typical phononic band structure (*f*–k-dispersion relation) of the phononic plate with a=15 mm, $w_{1}=6$ mm, $h_{1}=0.75$ mm, $w_{2}=1.05$ mm, and $h_{2}=6$ mm. (**a**) The calculated band structure without filtering. The two branches with high slopes are the $S_{0}$/$SH_{0}$ modes, and the rest are the $A_{0}$ flexural modes. (**b**) The filtered band structure with only flexural modes, with the upper and lower bounds indicated by the red and green bars, respectively.

**Figure 6 materials-16-01879-f006:**
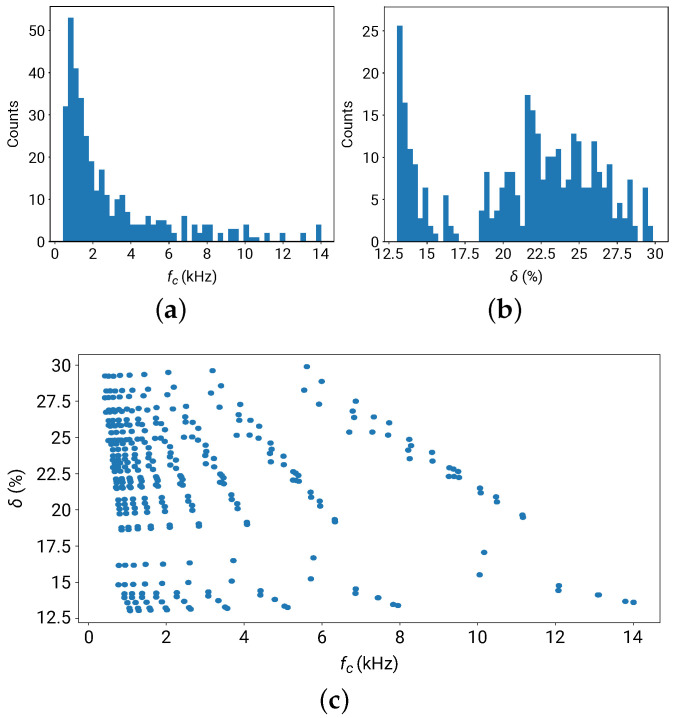
Distribution of the band gap frequency in the generated dataset. (**a**) Distribution of the center frequency $f_{c}$. (**b**) Distribution of the normalized half-bandwidth δ. (**c**) Scatter plot of $\left(f_{c},\delta \right)$.

**Figure 7 materials-16-01879-f007:**
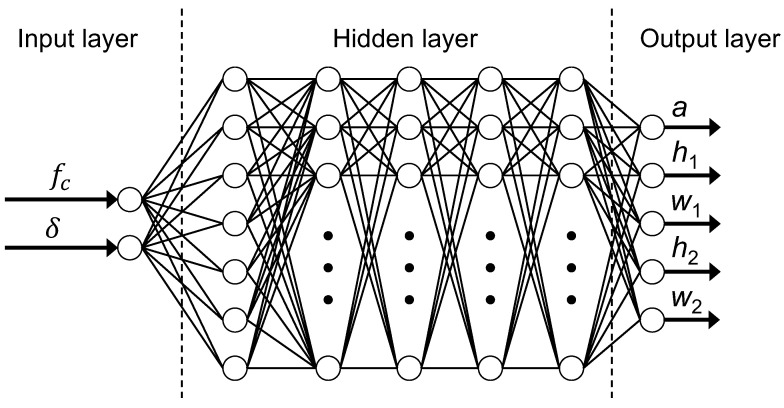
Diagram of the DNN architecture used in this study.

**Figure 8 materials-16-01879-f008:**
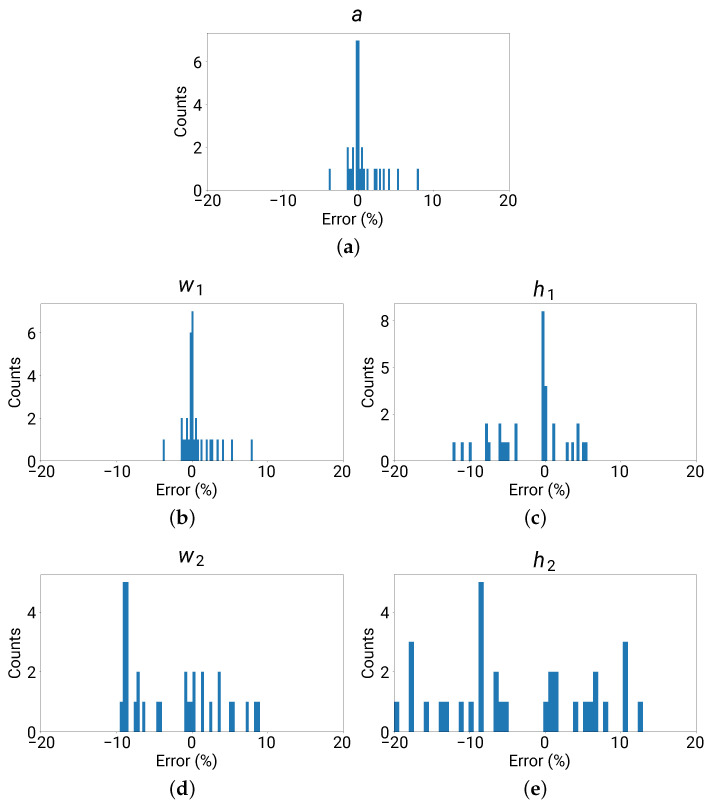
Error distribution for each of the five parameters of the 33 sets of testing data. (**a**–**e**) plot the error distributions for the parameters *a*, $w_{1}$, $h_{1}$, $w_{2}$, and $h_{2}$, respectively. The error metrics in Equation (Equation 7) were adopted.

**Figure 9 materials-16-01879-f009:**
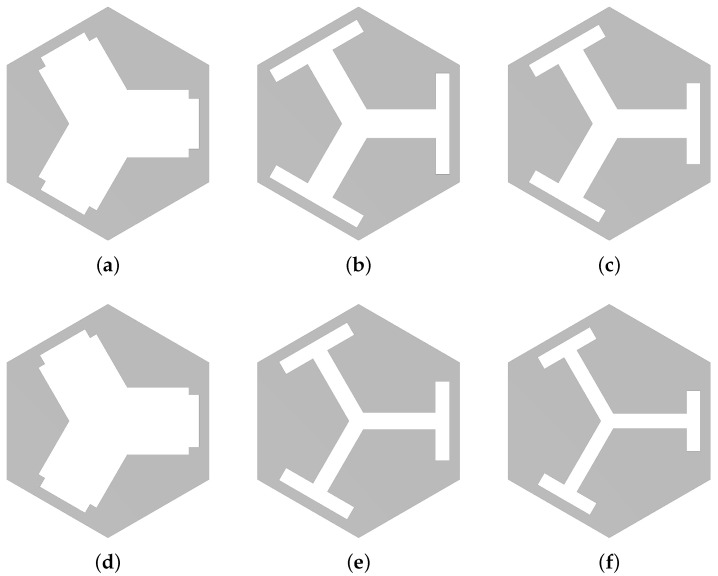
The shapes of the ground-truth (**a**–**c**) unit cells from the dataset and the predicted unit cells (**d**–**f**), for the three case studies. (**a**,**d**) Case 1. (**b**,**e**) Case 2. (**c**,**f**) Case 3. Note that they were scaled to the same size in the figures, while they had different lattice constants.

**Figure 10 materials-16-01879-f010:**
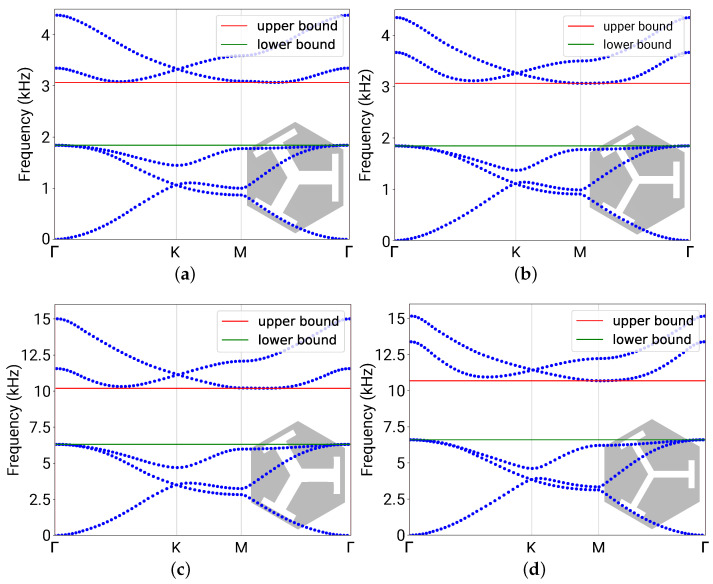
Band structures for the phononic plates considered in the case studies. (**a**) The prepared (ground-truth) phononic plate in Case 2. (**b**) The DNN-predicted phononic plate in Case 2. (**c**) The prepared (ground-truth) phononic plate in Case 3. (**d**) The DNN-predicted phononic plate in Case 3.

**Figure 11 materials-16-01879-f011:**
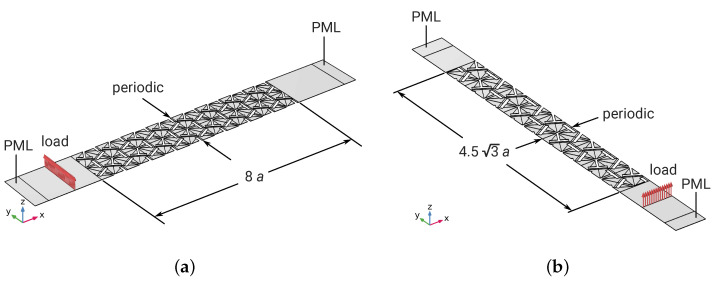
Finite-element models for the metamaterial transmission loss numerical experiments. (**a**) Transmission along the Γ-K direction. (**b**) Transmission along the Γ-M direction.

**Figure 12 materials-16-01879-f012:**
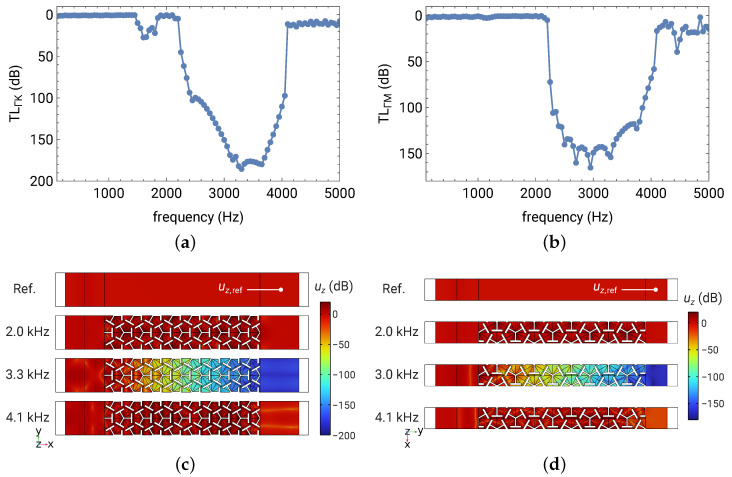
Transmission spectra and visualization of the designed metamaterial. (**a**) Transmission loss of flexural waves along Γ-K direction through the metamaterial of 8a length. (**b**) Transmission loss along Γ-M direction through the metamaterial of $4.5\sqrt{3}a$ length. (**c**) Visualization of flexural wave attenuation along Γ-K direction by plotting the $u_{z}$-level in and outside the band gap. (**d**) The $u_{z}$-level for flexural waves propagating in Γ-M direction.

**Table 1 materials-16-01879-t001:** Errors of the DNN predictions for the three case studies discussed in Section 4.2. For the error in the design parameters, the normalized mean absolute error evaluated using Equation (Equation 7) was used. Normalized absolute errors were used for the band gap ($f_{c}$ and δ).

Case #	Mean Error in Design Parameters	Error in $f_{c}$	Error in δ
1	1.25%	0.13%	0.98%
2	7.44%	0.18%	0.69%
3	6.47%	4.71%	0.37%

**Table 2 materials-16-01879-t002:** Baseline comparison between the proposed deep neural network (DNN) approach and four alternative machine learning algorithms. The performance of each algorithm was evaluated by the errors in $f_{c}$ and δ comparing the obtained band gap (forward calculation using the design parameters output from each algorithm) with the desired band gap. (${}^{*}$ denotes the proposed approach.)

Algorithm	Error in $f_{c}$	Error in δ
DNN ${}^{*}$	2.26%	1.75%
SVR	12.5%	6.91%
RFR	4.04%	1.60%
XGB	4.08%	1.76%
KNN	2.47%	2.04%

**Table 3 materials-16-01879-t003:** The hyperparameters used for the baseline approaches.

Algorithm	Hyperparameter	Description	Value/Function
	kernel type	Kernel function	RBF
SVR	C	Penalty parameter	80
	gamma	Kernel coefficient	5
RFR	criterion	Function to measure the quality of a split	absolute_error
	max_depth	Max. depth of the tree	40
	max_features	# of features to consider when looking for the best split	sqrt
	min_samples_leaf	Min. # of samples required to be at a leaf node	1
	min_samples_split	Min. # of samples required to split an internal node	3
	n_estimators	Number of trees in the forest	130
	objective	Learning objective	reg:squarederror
	max_depth	Max. depth of the tree	17
	min_child_weight	Min. sum of instance weight	3
XGB	colsample_bytree	Subsample ratio of columns	1
	eta	Learning rate	0.15
	subsample	Subsample ratio of training instances	0.5
	n_estimators	Number of trees in XGBoost	170
	weights	Weight function used in prediction	distance
KNN	p	Power parameter for the Minkowski metric	1
	n_neighbors	Number of neighbors	3

**Table 4 materials-16-01879-t004:** Results of baseline comparison of the training time, memory demand, prediction time, and number of parameters. (${}^{*}$ denotes the proposed approach.)

Algorithm	Training Time (s)	Peak Memory during Training (MB)	Prediction Time (s)	Peak Memory during Prediction (MB)	Number of Parameters
DNN ${}^{*}$	298.9139	0.9641	0.0019	0.0130	5097
SVR	0.0589	0.0434	0.0049	0.0181	3
RFR	1.1470	0.1987	0.0288	0.0438	6
XGB	0.4008	0.0904	0.0043	0.0226	7
KNN	0.0027	0.0215	0.0041	0.0181	3

## Data Availability

The proposed approaches are implemented in COMSOL Multiphysics and Python. All the simulation datasets and the algorithms introduced in this study should be able to reproduce the results with the details provided in Section 2 and Section 3. The corresponding author may be contacted if there are additional queries for implementations.

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
