# Peer review of "Deep-Learning-Based Acoustic Metamaterial Design for Attenuating Structure-Borne Noise in Auditory Frequency Bands"

_materials, 2023, doi:10.3390/ma16051879_

Round 1

Reviewer 1 Report

The current study shows promising results for the acoustic metamaterials research field.

However, some points should be taken into consideration:

-          May you explain what are w1 , h1 , w2, h2 label

-          In addition to finite element (FE) methods, do you think that the TMM method can be applied here

-          In the last two figures in the following study Fig 14 and 15, they studied the x-axis with wavenumber only to study the acoustic behavior of some materials, but in your study is considered with reduced wavenumber. What are the differences? I see that Fig 5 in your study is should be re-plotted again

Terahertz resonance frequency through ethylene glycol phononic multichannel sensing via 2D MoS2/PtSe2 structure - ScienceDirect

-          What is the software that has been used to study the Band structures in your study, is it also comsol?

-          For figure 12 a, b, I see that under both the electrical and magnetic mods, there are no big differences in your results. May you show the reasons behind

-          Before the conclusion section, it is better to add a small table explaining the differences between your results and the current literature review

Reviewer 2 Report

Authors reported a deep-learning based model for designing phononic plate structure. Mindlin theory of  plates  was used for modeling the pate and the neural network was trained for inverse design. The paper is of scientific importance to the computational material community and can be considered for publication after incorporating the following minor points in the revised manuscript.

1.      In this work, the band gap characteristics are considered. Usually, band gap frequencies denote the region in which the vibration and elastic wave cannot propagate. However, a recent result shows that besides this isolation property, the band gap also corresponds to the fracture resistance and meta-arrest characteristics, which can be found in the  article [Meta-arrest of a fast propagating crack in elastic wave metamaterials with local resonators. Mechanics of Materials 148, 103497] 

This work should mention that besides the wave isolation in the band gap, the elastic/acoustic wave metamaterials with local resonators has demonstrated its crack arrest and fracture resistance in the frequency region.

2.      Authors should  provide details about implenetation of the periodic boundary conditions in the FE formulation for extracting band structures.

3.      The underlying physical mechanism of the band gap should be further discussed.

4.      Add closely related recent articles on band gap structures in the ref. list.

(a)   "Functionally Graded Soft Dielectric Elastomer Phononic Crystals: Finite Deformation, electro-elastic longitudinal waves, and band gaps tunability via electro-mechanical loading" International Journal of Applied Mechanics, 14(5), 2250050 (2022) .

(b)   "Gradient-based topology optimization of soft dielectrics as tunable phononic crystals" Composite Structures 280, 114846 (2022).

(c)   "Topology Optimization of Soft Compressible Phononic Laminates for Widening the Mechanically Tunable Band Gaps" Composite Structures 289, 115389 (2022).

Reviewer 3 Report

Liu et al. have presented the manuscript titled: AI-based Acoustic Metamaterial Design for Attenuating Structure-borne Noise in Auditory Frequency Bands. Overall presentation of the article is good, and authors have provided the detailed study. I just have few suggestions for the authors about this article.

1.      Is it possible to use DNN (Deep Neural Network) like Multi-Layer Perceptron’s (MLP), Convolutional Neural Networks (CNN) or Recurrent Neural Networks (RNN) to attain the results of baseline comparison for prediction the band gap center frequency fc and the half-bandwidth δ closer values in Table 2?

2.      Can K-Nearest Neighbors Algorithm (KNN) be used except Support Vector Regression for the Error comparison due to high fluctuations in comparative results because of prediction discrete values best fit line is the hyperplane?

3.       It is batter to move your conditional operator arrow (in case of NO) towards NN input band gap to retrain with new parameters of Forward calculation matched w/ band gap in Figure 3.

4.      Title can be narrower by using Neural Network approach in place of AI-based because Artificial Intelligence have broader meanings.

Reviewer 4 Report

A few major issues need to be resolved before this manuscript can be accepted.

1)      The abstract should be revised as it does not enough chiefly introduce the area of research along with the research question.

2)      The research gap is not defined.

3)      Time series analysis is a challenging task. However, a discussion on it is missing in the introduction and literature section. Following are recent papers related to time series analysis: (https://doi.org/10.3390/math9243326, https://doi.org/10.1016/j.buildenv.2022.109531)

4)      The presentation of the paper is not good.

5)      Motivation is not clear.

6)      I suggest the authors write their main contributions in bullets.

7)      I suggest the authors use some recent review papers to summarize the state-of-the-art, discuss the main challenges, and compared their results with the proposed approach.

8)      It is critical to point out how to set the hyper-parameters of the machine learning methods. how can we know that the tuning of the parameter will not affect the accuracy of the methods?

9)      I suggest the authors provide the comparative complexity analysis of their proposed method.

10)  The language is poor and needs polishing.

Round 2

Reviewer 1 Report

The authors did the required revisions, thank you 

Reviewer 2 Report

Accept.

Reviewer 4 Report

The paper is revised significantly.